# Extrusion of process flavorings from methionine and dextrose using modified starch as a carrier

**Sirinapa Sasanam**[1], **Benjawan Thumthanaruk**[1], **Sombat Wijuntamook**[2], **Vasan Rattananupap**[2], **Savitri Vatanyoopaisarn**[1], **Chureerat Puttanlek**[3], **Dudsadee Uttapap**[4], **Solange I. Mussatto**[5], **Vilai Rungsardthong**[1] *

**1** Department of Agro-Industrial, Food and Environmental Technology, Faculty of Applied Science, Food and Agro-Industrial Research Center, King Mongkut's University of Technology North Bangkok, Bangsue, Bangkok, Thailand, **2** Mighty International Co., Ltd., Ladprao, Bangkok, Thailand, **3** Department of Biotechnology, Faculty of Engineering and Industrial Technology, Silpakorn University, Nakhon Pathom, Thailand, **4** Division of Biochemical Technology, School of Bioresources and Technology, King Mongkut's University of Technology Thonburi, Bangkhuntian, Bangkok, Thailand, **5** Department of Biotechnology and Biomedicine, Technical University of Denmark, Kongens Lyngby, Denmark

* vilai.r@sci.kmutnb.ac.th

**Data Availability Statement:** Data are available from the https://figshare.com/account/home#/data. There are the link to access as follows

## Abstract

This study aimed to produce process flavorings from methionine and glucose via Maillard reaction by extrusion method. Modified starch was used as a carrier to reduce the torque and facilitate the production process. Five formulations of process flavorings with different ratios of methionine: dextrose: modified starch: water as MS5 (72:18:5:5), MS15 (64:16:15:5), MS25 (56:14:25:5), MS35 (42:12:35:5), and MS45 (40:10:45:5) were prepared and feded into the extruder. The temperatures of the extruder barrel in zones 1 and 2 were controlled at 100, and 120°C, with a screw speed of 30 rpm. The appearance of the obtained products, torque, pH before and after extrusion, color, volatile compounds, and sensory evaluation were determined. The extrudate from the formulation containing the highest amount of modified starch (MS45) gave the highest L* (lightness) of 88.00, which increased to 93.00 (very light) after grinding into a powder. The process flavorings from all formulations exhibited similar sensory scores in terms of aroma, taste, and water solubility, with a very slight difference in color. However, MS25, MS35 and MS45 indicated the torque at 10 Nm/cm$^3$, while MS5 and MS 15 exhibited higher torque at 18, and 25 Nm/cm$^3$, respectively. Extruded process flavorings from MS25 were analyzed for their flavor profiles by gas chromatography-mass spectrometry. Twelve volatile compounds including the key volatile compounds for sulfurous and vegetable odor type, dimethyl disulfide, methional, and methanethiol, were found. Four pyrazine compounds presented nutty, musty and caramelly odor; and 3-hydroxybutan-2-one and heptane-2,3-dione, which gave buttery odor type, were also detected. The results demonstrated a successful production of process flavorings using modified starch as carrier to facilitate and reduce the torque during the extrusion process.

Manuscript1: Body texts: https://doi.org/10.6084/m9.figshare.19608813 Manuscript2: Tables: https://doi.org/10.6084/m9.figshare.19608816 Manuscript2: Tables: https://doi.org/10.6084/m9.figshare.19608816 Fig 1. https://doi.org/10.6084/m9.figshare.19608681 Fig 2. https://doi.org/10.6084/m9.figshare.19608687 Fig 3. https://doi.org/10.6084/m9.figshare.19608699 Table 4. https://doi.org/10.6084/m9.figshare.19608735 Supporting information. https://doi.org/10.6084/m9.figshare.19608747.

**Funding:** The work was funded by Research and Researchers for Industries (RRI) scholarship (PHD60I0062) and co-funded with Mighty International Co., Ltd., and Novo Nordisk Foundation, Denmark (grant number NNF20SA0066233). The funders had no role in study design, data collection and analysis, decision to publish, or preparation of the manuscript. There is clarification the sources of funding as follows a) Please clarify the sources of funding (financial or material support) for your study. List the grants or organizations that supported your study, including funding received from your institution. Ans: • Research and Researchers for Industries (RRI) scholarship (PHD60I0062) supported the finances for this study. • Mighty International Co., Ltd., supported the finances and materials for this study. • Prof. Solange I. Mussatto acknowledges the support from the Novo Nordisk Foundation, Denmark (grant number NNF20SA0066233). b) State what role the funders took in the study. If the funders had no role in your study, please state: "The funders had no role in study design, data collection and analysis, decision to publish, or preparation of the manuscript." Ans: The funders had no role in study design, data collection and analysis, decision to publish, or preparation of the manuscript." c) If any authors received a salary from any of your funders, please state which authors and which funders. Ans: The author "Sirinapa Sasanam" received a salary from Research and Researchers for Industries (RRI) scholarship (PHD60I0062) during the study. d) If you did not receive any funding for this study, please state: "The authors received no specific funding for this work." Ans: "The authors received no specific funding for this work."

**Competing interests:** NO authors have competing interests.

## Introduction

The combination of tender, juicy meat and an intense meat flavor is essential for good quality consumption. Meat flavor is a very important factor for consumers in the selection of meat products [1, 2]. There are many types of meaty flavors used in food industries, such as grilled chicken, beef, cheese, egg salted, seafood and shrimp flavors. Precursors in the raw meat and heating process are the main factors for the generation of specific meaty volatile compounds [3–5].

The volatile flavor compounds such as aldehydes, pyrazines, ketones, hydrocarbons, alcohols, nitrogen and sulfur-containing compounds are organic compounds in nature which have low molecular weight and low odor threshold [6]. Many volatile compounds including 2-ethyl-3-furanthiol, 2-methyl-3-(methyldithio) furan, 3-mercapto-2-methyl-1-pentanol can present meaty odor [7, 8]. The highly odor active compounds and reactive intermediates are generated from different amino acids in the Maillard reaction, such as phenylacetaldehyde from phenylalanine, which gives honey and rose odor with an odor threshold at 5.2 μg/kg, and methional from methionine exhibited cooked potato aroma at 0.4 μg/kg [9]. At the same time, many reactive intermediates compounds such as formaldehyde from glycine, acetaldehyde from alanine and cysteine can also generate specific volatile compounds [9].

Maillard reaction is a non-enzymatic browning reaction that occurs during food processing, being related to the generation of various flavoring and volatile compounds. The reaction involves the loss of a water molecule between amine groups of amino acids and carbonyl groups of aldehydes or ketones of reducing sugars [10, 11]. The Maillard reaction, as a natural modifying method, is often used to improve the functional properties of proteins or polysaccharides [12].

Reflux process in a liquid system is widely used for the production of process flavorings in the conventional manufacturing method. May [13] and Reineccius [14] reported the production of beef process flavorings by heating under the reflux system using a ratio of mixed substrates from cysteine hydrochloride: wheat protein hydrolysate liquid 40%: D-xylose: water of 2:24:2:72, for 3 h and cooled within 30 min. The samples after reaction were further added with a carrier at the ratio of 60:40 and dried into powder using a spray dryer. This method is a long and complex process which consumes and produces a large amount of influent and effluent. Alternative methods, more environmentally friendly and with higher efficiency, technologies are required for the production of meaty process flavorings.

Extrusion process is a potential modern, green and environment-friendly technology that combines several processing steps such as mixing, cooking, sterilization, heating, and forming. Some advantages of this process include versatility, low cost, high-temperature short time (HTST), and high efficiency in food processing [15]. The extrusion is more cost-effective to operate than the traditional heating systems because it performs multiple unit operations in one machine, resulting in higher productivity and lower production costs when compared to conventional methods [16]. Many chemicals and structural transformations such as protein denaturation, complex formation between amylose and lipids, degradation reactions of vitamins or pigments starch gelatinization could occur during the extrusion process [17, 18]. The low moisture content of the initial feed, high temperature, and short-time processing in the extruder results in the formation of several volatile compounds via Maillard reaction. The specific flavors found depend on the type of nitrogen sources or amino acid and reducing sugar used for the reaction [19].

Compared to hexoses, pentose sugars races with amino acid faster and are more reactive to generate Maillard reaction. Sasanam et al. [20] found that the use of D-xylose with methionine exhibited camphor odor from the 1,7,7-trimethylbicyclo [2.2.1] heptan-2-one, while using

dextrose as the reducing sugar presented sweet, creamy, caramelly, buttery with a fruity jammy nuance odor description and buttery odor type from the ketone compound, namely, 2,3-hexanedione. The authors also reported the extrusion of four model reactions from methionine, thiamine, and D-xylose or glucose by direct extrusion without using any carrier. Though potential process flavorings were successfully obtained, the extrusion output was low and not easy to control since a very high torque (around 85–117 Nm/cm$^3$) occurred during the extrusion.

The use of a carrier to facilitate the extrusion of food products has been reported in some studies [21–23]. However, there are only a few studies reporting the use of carrier for the extrusion of process flavorings [19, 24]. A smooth and easy to control extrusion of beef process flavoring was feasible when wheat starch was applied to the mixed substrates [25]. However, to the best of our knowledge, there are no reports on the use of modified cassava starch as a carrier to solve the problems related to the high torque, too high viscosity, and stickiness of the raw materials to produce process flavorings by extrusion. Thailand is one of the top countries for cassava cultivation and processing as several value-added products such as cassava flour, cassava starch, and modified cassava starch, both for local consumption and exporting. Therefore, modified cassava starch was used to facilitate and obtain a better control of the extrusion of process flavorings from methionine and dextrose in the present study. The torque used during the extrusion, as well as the properties of the mixed substrates and the extruded process flavorings were determined. The flavorings from each formulation were evaluated for their properties and sensory evaluation. The volatile profiles of the product from a selected formulation were analyzed by gas chromatography-mass spectrometry (GC-MS). The information generated in this study will be useful for industrial scale up of process flavorings production by extrusion in the future.

## Materials and methods

### Materials

The raw materials used for the extrusion of process flavorings were kindly obtained from Mighty International Co., Ltd., and included methionine (99%), dextrose monohydrate (99%), and a modified starch (cross-linked type). The standard n-alkanes (C5–C19) (chromatographic reagent) were supplied from Sigma—Aldrich (Santa Clara, USA).

### Process flavorings production by extrusion

The formulation of methionine and reducing sugar, which exhibited meaty process flavorings, was selected based on the literature [26]. Dextrose was used in this study instead of D-xylose since the use of D-xylose in our previous study [20] resulted in a camphor odor that might not be well accepted as meaty process flavorings. Five formulations of methionine: glucose: modified starch: water, consisted of modified starch at 5, 15, 25, 35, and 45% (by weight): MS5 (72:18:5:5), MS15 (64:16:15:5), MS25 (56:14:25:5), MS35 (42:12:35:5), and MS45 (40:10:45:5), were prepared. The substrates were mixed in a mixer (Kenwood KM230, England) for 25 min, and then fed into a single screw extruder (Brabender 19/20DN, Germany with barrel length of 386 mm and 19.1 mm screw diameter). The temperatures of the barrel in zones 1 and 2 were operated at 100 and 120˚C, with a screw speed of 30 rpm. The torque during production was measured using a sensor with a monitor. The extrudates were ground into powder using a hammer mill (BOSCH MCM3501M, Germany) and passed through a sieve (60 mm mesh size). The obtained powders were kept at -4˚C in brown glass bottles until further analysis and sensory evaluation. The steps to produce process flavorings by extrusion and overall experiments are presented in Fig 1.

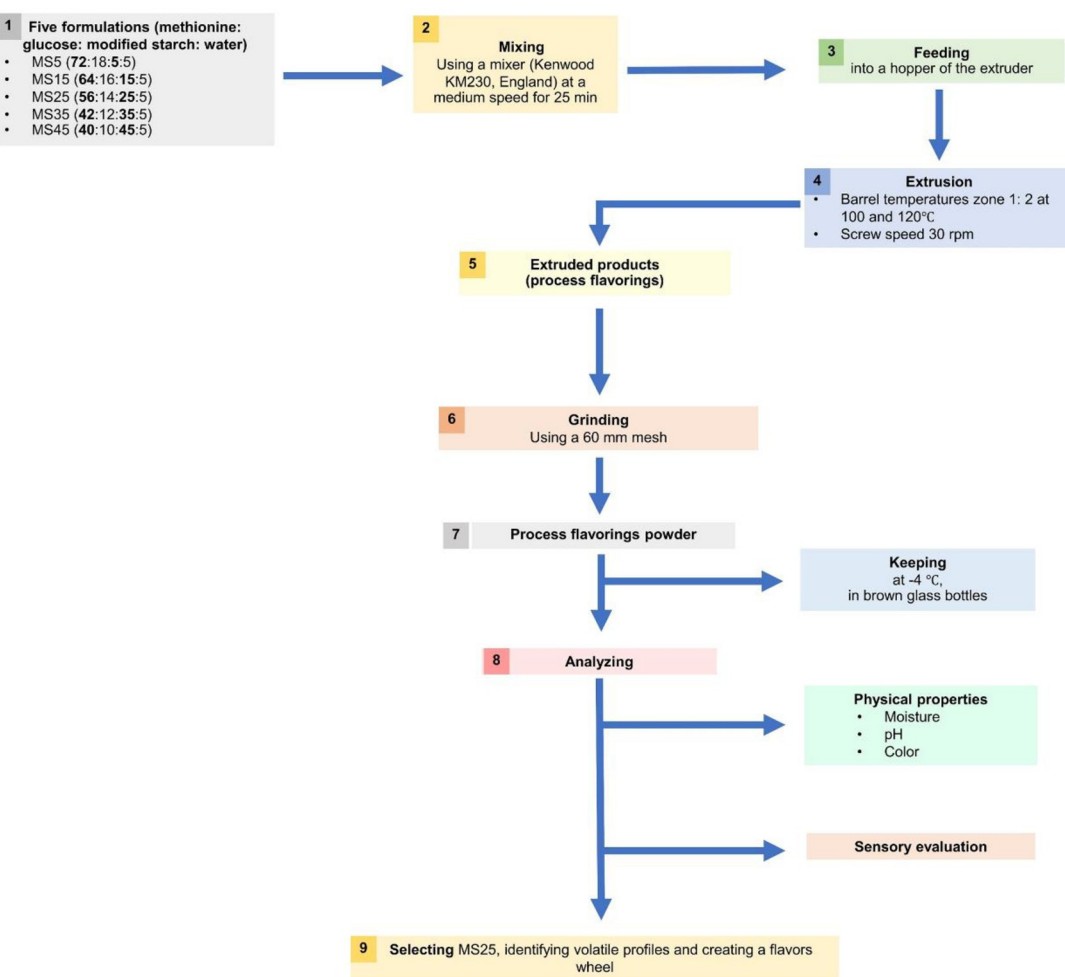

**Fig 1. Overall experiments for the study of process flavorings production by extrusion using modified starch as a carrier.**

### Properties of the extruded process flavorings

**Moisture and pH.** The moisture content and pH of the mixed substrates and obtained process flavorings were determined following the Association of Official Analytical Chemists [27]. The pH meter (Model 510, Cyberscan, Netherland) was used to measure pH of the products. The samples were suspended in distilled water (pH 7.3) at 1% (w/v) and mixed by a stirrer for 30 min before measurement.

**Color.** Color analysis was performed using a colorimeter (Color Quest XE, USA) with CIE Lab system. The considered parameters were $L^*$ (lightness on a 0 to 100 scale from black and white); $a^*$ (+redness/-greenness); and $b^*$ (+yellowness and -blueness). The browning index (BI) of each process flavorings was calculated based on the CIE values of $L^*$, $a^*$, and $b^*$ [28] as presented in the following equation.

BI = 100 x (X– 0.31)/ 0.17

Where, $X = (a^* + 1.75\ L^*) / (5.645L^* + a^* - 3.012b^*)$

**Analysis of volatile compounds by GC-MS.** The volatile compounds of the extruded process flavorings were analyzed by a GC-MS system (5977B MSD; Agilent Technologies Inc., Santa Clara, CA, USA) using a headspace solid-phase microextraction (HS-SPME) with a Teflon-lined septa, 50/30 μm divinylbenzene/ carboxene/ polydimethylsiloxane (DVB/CAR/

PDMS) fiber (Supelco Inc., Bellefonte, PA). The GC capillary column, HP-5MS (30 m × 0.25 mm i.d. × 0.25 μm film thickness), was operated at an extraction time of 60°C, for 40 min before the injection. The GC oven temperature was maintained initially at 40°C for 5 min and then increased to 250°C at the ratio of 4°C/min. The injector temperature was set at 250°C with desorption time of 5 min and a split mode of 1:1. The MS source temperature, electron-impact mass spectra, and scan range were set following methods described previously [26].

**Identification of the volatile compounds.** Volatile compounds were identified according to [20, 26, 29]. The mass spectrum and retention index with reference compounds were used for identification of volatile compounds using NIST 14 (National Institute of Standard and Technology) mass spectrum libraries. The series of standard alkanes including C5-C19, which were analyzed under the same GC-MS running condition, were used as references. The odor description of each volatile compound was referenced and reported based on the database of the website http://www.thegoodscentscompany.com/.

**Sensory evaluation.** Fifteen trained panelists from the Department of Research and Development, Mighty International Co., Ltd., with 9 females and 6 males (25–50 years), were asked to score the extruded process flavorings for the sensory test (1 = the least acceptable to 9 = the most acceptable scale). The samples from the process flavorings were prepared at 1% solution (w/v) by dissolving them in boiling water (around 100°C). The codes with 3-digit numbers were used for random labeling orders. The panelists were asked to rate the samples based on the aroma, taste, color, water solubility, and overall acceptability. The samples (~20 mL) with random codes were served in a plastic glass covered with aluminum foil and kept warm at 60°C. Panelists were instructed to drink some water at each sample's intervals, and took a break for a few minutes before the next samples were served. The evaluation was a blind test and no information about the components of the formulation was provided to the panelists [30]. Five-basic tastes, including citric acid (sour), sucrose (sweet), sodium chloride (salty), caffeine (bitter), and monosodium glutamate (umami) were used for training the panelists [31].

**Statistical analysis.** One way analysis of variance (ANOVA) was carried out to compare the mean by Duncan's multiple range test (DMRT) at a significance level of $p < 0.05$ using SPSS 17 for Windows (SPSS Inc., Chicago, III, USA).

## Results and discussion

### Properties of the mixed substrates and extruded process flavorings, and torque during the extrusion

The pH of mixed substrates, torque during extrusion, and some properties of the extruded process flavorings from each formulation are presented in Table 1. Exhibited torque during the extrusion of all five formulations ranged 10–25 Nm/cm³. Using modified starch at 5% by weight (MS5) presented the highest torque of 25 Nm/cm³, while an increase of the modified starch to 25–45% (by weight) could reduce the torque to 10 Nm/cm³. Sasanam et al. [20] reported that the torque of process flavorings from the model reaction of methionine: D-xylose, methionine: dextrose, thiamine: D-xylose, and thiamine: dextrose at the ratio of 80:20 by weight was as high as 85–117 Nm/cm³. The lower torque occurred during the extrusion led to a more continuously and smoothly controllable process with a higher output of the products reported. A better process control by the addition of modified wheat starch as a carrier to extrude the beef process flavorings by direct extrusion was reported in our previous study [25]. The obtained beef process flavoring presented comparable properties to a commercial beef process flavorings prepared by the traditional method (boiling and reflux method). However, the addition of modified starch reduced the viscosity of the mixed substrates, thus reducing the torque inside the extruder.

**Table 1. pH of mixed substrate and torque used during extrusion, as well as the appearance, pH and moisture content of extruded process flavorings produced from various ratios of substrates.**

| Sample | Methionine: dextrose: modified starch: water | pH | Torque (Nm/cm³) during extrusion | Extruded process flavorings | |
|---|---|---|---|---|---|
| | | | | pH | Moisture content (%) |
| MS5 | 72:18:5:5 | $6.37 \pm 0.01^d$ | 25 | $5.74 \pm 0.01^d$ | $5.33 \pm 0.38^b$ |
| MS15 | 64:16:15:5 | $6.41 \pm 0.02^c$ | 18 | $5.77 \pm 0.02^d$ | $5.43 \pm 0.26^b$ |
| MS25 | 56:14:25:5 | $6.42 \pm 0.01^c$ | 10 | $5.84 \pm 0.01^c$ | $4.64 \pm 1.10^c$ |
| MS35 | 42:12:35:5 | $6.45 \pm 0.01^b$ | 10 | $5.98 \pm 0.01^b$ | $5.01 \pm 1.40^c$ |
| MS45 | 40:10:45:5 | $6.50 \pm 0.01^a$ | 10 | $6.32 \pm 0.02^a$ | $6.88 \pm 0.40^a$ |

The barrel temperatures of zones 1, 2 were controlled at 100 and 120˚C with screw speed of 30 rpm.

Means ± SD in the same column, followed by the same letter, are not significantly different at $p < 0.05$.

The produced process flavorings exhibited very low moisture content, ranging from 4.64 to 6.88%, and the products from all formulations could be easily grounded into powder. No further drying was required. Consequently, the short and continuous extrusion was considered an efficient alternative process for the production of process flavorings compared to the conventional reflux method.

Mixed substrates containing 40% methionine, 10% dextrose and 45% of modified starch (MS45) presented the highest pH of 6.50, which decreased to 6.32 after the extrusion. In contrast, the mixed substrates from methionine at 72% with modified starch at 5% (MS5) showed the highest pH at 6.37, which decreased to 5.74 when extruded. The concentration of methionine and dextrose affected the reaction rate. It might be possible that during the extrusion process, amino groups from the methionine in the initial stage had depleted and generated organic acids in the final stage of the Maillard reaction [20, 32]. The reduced pH in the obtained product after extrusion process related to the Maillard reaction in the reaction liquid. The properties of the process flavorings obtained including pH and color, were significantly dependent on the concentration of precursors in the mixture. However, the role of the amino acid in the Maillard reaction could be separated into two: the first one is to promote a sugar-amino condensation, while the second one is to generate specific aromas via the Strecker degradation [26]. Different initial precursors, including amino acids and reducing sugars react at different rates depending on the type and concentration of the substrates. Some amino acids are more effective to generate the reaction in the early stage of the Maillard reaction, whereas others may be more effective during later stages [33].

Color is one of the indicators of the extent of Maillard reaction or the formation of melanoidin, which is generated from the cross-linking of peptides or free amino acids with reducing sugar. The condensation or loss of water molecules in the Maillard reaction leads to the formation of brown high-molecular-weight compounds [34]. The color of the mixed substrates and extruded process flavorings obtained in the present study is shown in Table 2. The products from MS45 gave the highest L* (lightness) of 88.00, which increased to 93.00 (very light) after grinding. On the other hand, using the modified starch at 25% or MS25, resulted in the product with the lowest L* of 68.00 (dark), which was increased to 87.00 when powdered. This result is in agreement with Sasanam et al. [20] who reported a L* value for the product obtained from methionine with dextrose at the ratio of 80:20 by weight equal to 85.37. Using high ratio of the modified starch from 5% to 45% led to a decrease in the concentration of methionine and dextrose which were the initial precursors for the Maillard reaction in the system. The lightness of the powder was increased due to the higher light scattering when the particle size of the process flavorings was decreased. In fact, the optical properties of the powder

**Table 2. Color indicators ($L^*$, $a^*$, $b^*$) of the extruded process flavorings produced from various ratios of substrates, before and after size reduction into powder.**

| Sample | Methionine: dextrose: modified starch: water | Extrudates | | | | Powdered extrudates | | | |
|---|---|---|---|---|---|---|---|---|---|
| | | $L^*$ | $a^*$ | $b^*$ | BI[1] | $L^*$ | $a^*$ | $b^*$ | BI[2] |
| MS5 | 72:18:5:5 | 79.33 ± 2.51[a] | 6.00 ± 0.00[c] | 33.33 ± 2.51[bc] | 60.09 ± 5.11[d] | 89.00 ± 2.64[ab] | 3.00 ± 0.00[a] | 24.66 ± 0.57[b] | 34.26 ± 0.89[b] |
| MS15 | 64:16:15:5 | 71.66 ± 4.50[b] | 6.00 ± 2.00[a] | 36.00 ± 2.64[ab] | 80.19 ± 14.22[b] | 90.33 ± 3.05[ab] | 1.66 ± 0.57[bc] | 28.33 ± 1.52[a] | 38.12 ± 1.88[a] |
| MS25 | 56:14:25:5 | 68.00 ± 5.00[b] | 9.33 ± 2.08[ab] | 34.00 ± 1.00[b] | 99.12 ± 42.76[a] | 87.00 ± 2.00[b] | 1.00 ± 0.00[d] | 22.00 ± 1.00[c] | 29.36 ± 2.03[c] |
| MS35 | 42:12:35:5 | 83.00 ± 6.92[a] | 6.66 ± 0.57[bc] | 28.00 ± 2.00[d] | 55.96 ± 8.43[e] | 87.66 ± 3.05[ab] | 2.33 ± 0.57[ab] | 9.66 ± 0.57[d] | 13.38 ± 1.28[d] |
| MS45 | 40:10:45:5 | 88.00 ± 2.14[a] | 7.33 ± 1.52[abc] | 37.00 ± 4.00[a] | 73.64 ± 22.55[c] | 93.00 ± 2.57[a] | 1.00 ± 1.00[cd] | 9.33 ± 0.57[d] | 11.92 ± 1.68[d] |

Means ± SD in the same column, followed by the same letter, are not significantly different at $p < 0.05$.

BI[1] = Browning index of extrudates

BI[2] = Browning index of powdered extrudates

are dependent on the degree of light scattering [35]. The intensity of the reflected light changes due to the diffused reflection. The more the grinding, the stronger the intensity of the reflected light, so that the color of the powder changes [36]. The brownish color of the Maillard reaction under refluxing method has also been reported by Al-Baarri and Legowo [37]. According to these authors, the physical browning value and spectral analysis from 190 to 620 nm of the Maillard reaction product obtained from methionine and glucose prepared in a liquid system by heating at 50 ˚C for 24 h was changed. They found the development of browning compound from methionine with glucose at 420 nm, with absorbance at 1.218; while sugar alone only generated absorbance at 0.644 [37]. Therefore, color is an important factor to evidence the completion of the Maillard reaction.

The browning index (BI) is an important parameter to measure the brown color of process flavorings produced from the Maillard reaction [38]. The BI of each sample are presented in Table 2. After grinding, BI of all process flavorings tended to significantly decrease ($p < 0.05$). This could be due to the higher light scattering occurred when the extrudates were size reduced into finer particles. The BI of powdered extrudates of all formulations ranged 11.92–38.12. However, the BI of each sample did not correlate well with the sensory acceptability in term of color attribute as presented in Table 3. The sensory panelists preferred the product color from the samples extruded with the modified starch 15–45% than the use of 5% modified starch, but the difference was very slight. However, all samples from the modified starch 5–45% obtained non-significant difference for overall acceptability (5.53–6.00).

## Sensory evaluation of process flavorings

The sensory evaluation of process flavorings obtained from different formulations is presented in Table 3. Using modified starch at 5–45% (by weight) resulted in no significant differences

**Table 3. Sensory evaluation of process flavorings produced from various ratios of mixed substrates using a 9-point scoring scale.**

| Sample | Methionine: dextrose: modified starch: water | Aroma[ns] | Taste[ns] | Color | Water solubility[ns] | Overall acceptability[ns] |
|---|---|---|---|---|---|---|
| MS5 | 72:18:5:5 | 5.40 ± 2.35 | 5.13 ± 1.99 | 5.20 ± 1.74[b] | 6.60 ± 1.12 | 5.53 ± 1.76 |
| MS15 | 64:16:15:5 | 5.66 ± 1.67 | 5.06 ± 1.94 | 6.80 ± 1.37[a] | 6.80 ± 1.01 | 5.86 ± 1.64 |
| MS25 | 56:14:25:5 | 5.46 ± 1.68 | 6.20 ± 0.07 | 5.60 ± 1.68[ab] | 7.00 ± 0.84 | 6.00 ± 1.25 |
| MS35 | 42:12:35:5 | 5.26 ± 1.94 | 5.40 ± 1.95 | 6.00 ± 1.81[a] | 6.20 ± 1.26 | 5.66 ± 1.67 |
| MS45 | 40:10:45:5 | 5.00 ± 1.64 | 5.80 ± 1.20 | 5.60 ± 1.80[ab] | 7.00 ± 0.65 | 5.66 ± 1.44 |

The temperature in zones 1, 2 was controlled at 100, 120˚C with a screw speed of 30 rpm.

Means ± SD in the same column, followed by the same letter, are not significantly different at $p < 0.05$.

ns: *Not significantly different at $p < 0.05$*

($p<$ 0.05) in terms of aroma, taste, water solubility and overall acceptability. Though all process flavorings presented similar sensory score, the use of modified starch at 25–45% (MS25-MS45) could reduce the torque needed during extrusion from 18, and 25 Nm/cm$^3$ (for MS5 and MS15, respectively) to 10 Nm/cm$^3$. However, increasing the modified starch to 5 and 25% carrier (MS5-MS25) tended to receive higher score for overall acceptability and water solubility. The results showed that the addition of higher modified starch could facilitate the extrusion with lower torque, for example for the MS25 and MS45. Running the samples with higher ratio of the modified starch tended to exhibit lower torque during the extrusion and yielded samples with lower BI. However, both samples still obtained non-significant difference for color preference. Increasing the ratio of the modified starch resulted to a lower concentration of methionine and dextrose in the sample mix thus a lower concentration of the brown products from the Maillard reaction was generated. Moreover, the torque during production process had stabled with consistency level since using modified starch as carrier at 25%. It would be interested to select this sample for developing in term of taste and profile by adding another ingredient. Therefore, the sample MS25 was then used to identify the volatile as well as to create the flavor wheel to predict these flavor profile. Consequently, the extruded process flavorings from MS25 was selected for further studies on volatile analysis. The findings from our work would be potentially help the food industry to use the extrusion to produce process flavorings. The use of carriers including modified starches can facilitate the extrusion to be controlled more efficiently. In addition, modification of the type and concentration of sugars, amino acids, and carriers would be beneficial for further applications.

## Volatile compounds of meaty process flavorings

The sample MS25 was selected for the analysis of volatile compounds since this formula used the lowest concentration able to require low torque during the extrusion (10 Nm/cm$^3$). Twelve volatile compounds found in the process flavorings from MS25 are summarized in Table 4. Dimethyl disulfide and methional were the key meaty volatiles, which gave sulfurous and vegetable odor type. Sulfur-containing volatile compounds significantly contribute to meat aroma [39]. Many sulfur-containing compounds such as thiophenes, the sulfur, have been reported to exhibit a strong meaty flavor in meat products [40]. Sulfur compounds are widely used as flavorings for food products such as cheese, garlic, and meats [41]. Four pyrazine compounds were the main compounds (around 33%) identified by GC-MS. The compounds, 2-methyl-3-(2-methylpropyl) pyrazine, 2,3-diethyl-5-methylpyrazine, 3,5-diethyl-2-methylpyrazine and ethanone, 1-(3,5-dimethylpyrazinyl), present caramelly, musty and nutty odor type. It might be possible that many pyrazine compounds were generated under high temperatures of the extruder barrel in zones 1 and 2 at 100 and 120°C. This result was in accordance with other authors [42] who also reported an increase of pyrazine compounds such as methylpyrazine, 2,5-dimethylpyrazine, and 2,3-dimethylpyrazine when the temperature of the Maillard reaction of D-xylose and chicken peptide was increased from 100°C to 140°C. This result was also in accordance with [43] who reported that heterocyclic compounds such as pyrazines, pyridines, and thiazoles in the chicken meat were generated at 80–140°C. Pyrazine compounds were considered as the key aroma-active compounds that contributed to the aroma of Maillard reaction products [42].

Comparison of flavor profile of process flavorings between using modified starch as a carrier compared to process flavorings without carrier from our previous study. Adding the modified starch at 25% by weight led to a decreased concentration of the other compounds in the mixed substrates, especially amino acids and reducing sugar, which affected/ changed the volatile profile. Sasanam et al. [20] found fourteen volatile compounds in process flavorings

**Table 4. Volatile compounds, odor, and taste description of process flavorings extruded from methionine: dextrose: modified starch: water at the ratio of 56:14:25:5.**

| No | Volatile compounds | RT | Odor type | Odor description | Flavor type | Taste Description |
|---|---|---|---|---|---|---|
| *Strecker aldehyde (1)* | | | | | | |
| 1 | 2-methylpropanal | 1.841 | Aldehydic | Fresh, aldehydic, floral, pungent | Aldehydic | Fresh, aldehydic, herbal, green, malty |
| *Carboxylic acid compound (1)* | | | | | | |
| 2 | Acetic acid | 1.958 | Acidic | Sharp, pungent, sour, vinegar | Sour | Pungent, sour, fruit, overripe fruit, acetic |
| *Ketone compounds (2)* | | | | | | |
| 3 | 3-hydroxybutan-2-one | 3.227 | Buttery | Sweet, buttery, creamy, dairy, milky, fatty | Creamy | Creamy, dairy, sweet, oily, milky, buttery |
| 4 | Heptane-2,3-dione | 6.924 | Buttery | Butter, cheese, oily, fresh yogurt | Buttery | Sweet, dairy, butter, butterscotch, butter rum, caramel |
| *Ester compound (1)* | | | | | | |
| 5 | 2-oxopropyl acetate | 8.442 | Fruity | Fruity, buttery, dairy, nutty | * | * |
| *Pyrazine compounds (4)* | | | | | | |
| 6 | 2-methyl-3-(2-methylpropyl) pyrazine | 19.108 | Caramelly | Herbal, green, sugar, syrup | Green | Green, vegetable |
| 7 | 2,3-diethyl-5-methylpyrazine | 19.916 | Musty | Musty, nutty, meaty, vegetable, roasted, hazelnut | Musty | Musty, toasted, nutty, potato, cocoa, earthy and dirty |
| 8 | 3,5-diethyl-2-methylpyrazine | 19.995 | Nutty | Nutty, meaty, vegetable | Nutty | Green, nutty |
| 9 | Ethanone, 1-(3,5-dimethylpyrazinyl) | 22.291 | Nutty | Nutty, roasted, hazelnut | Nutty | Nutty, roasted, hazelnut |
| *Sulfide compounds (2)* | | | | | | |
| 10 | Dimethyl disulfide | 3.888 | Sulfurous | Sulfurous, vegetable, cabbage, onion | Sulfurous | Sulfurous, cabbage, malt, cream |
| 11 | Methional | 9.677 | Vegetable | Vegetable oil, Creamy tomato, potato skin and French fry, yeasty, bready, limburger cheese with a savory meaty brothy nuance | Tomato | Creamy tomato, potato skin and French fry, yeasty, bready, limburger cheese with a savory meaty brothy nuance |
| *Thiol compound (1)* | | | | | | |
| 12 | Methanethiol | 1.508 | Sulfurous | Vegetable oil, alliaceous, eggy, creamy, savory | Sulfurous | Sulfurous, alliaceous, creamy, cheesy, clean, savory, meaty |

* Means no odor was detected or no report on their odor description/ odor profile.

LRI and MS were used for compound identification.

produced from methionine with dextrose at 80:20 by weight, without using carrier. The authors also found three sulfide compounds, and some key volatile compounds that present sulfurous and vegetable odor type, namely, dimethyl disulfide, dimethyl trisulfide and methional, methanethiol, and 2-methyl-3-furanthiol, in their product. On the other hand, using modified starch as a carrier at 25% with methionine and dextrose at 56 and 14%, respectively, resulted in a smaller number of volatile compounds detected in the present study. Pyrazine compounds were the main volatile compounds instead of sulfide compounds and thiol compounds. Only dimethyl disulfide, methional and methanethiol were detected in the sample MS25 when a low concentration of methionine was used.

There are only a few studies reporting the volatile compounds present in process flavorings obtained by extrusion and by the conventional method. Sakunwaropat et al. [25] identified thirteen volatile compounds in beef process flavorings produced by extrusion. The authors also found methanethiol, a key component contributing to stewed beef and ground beef, as well as 2-furancarboxaldehyde, 3-(methylthiol)-propanal, 2- (methylthio) methyl-furan. Methanethiol gave stewed beef, ground beef odor in both, extruded products and commercial beef flavors. In fact, methanethiol is a key volatile compound found in beef flavors [44].

Three amino acids, glycine, alanine, or valine with D-xylose were used by Sasanam et al. [26] to produce process flavorings by extrusion process. The authors found thirteen, and twelve volatile compounds in the products obtained from D-xylose with glycine, and alanine, respectively, at the ratio of 20:80 by weight. Using valine, the higher molecular weight amino acid tested, thirty-seven volatile compounds with more complex flavor profiles were found, among of which, the compound 5-methyl-2-phenylhex-2-enal, which exhibits a chocolate odor, was also present.

Strecker aldehydes including 2-methylpropanal, which exhibits aldehydic odor type and fresh, aldehydic, floral, pungent odor, was also detected in the present study. This compound could have been generated during Maillard reaction. The condensation reaction between amine group of amino acid and carbonyl group of reducing sugar, leads to the generation of Amadori products or Amadori rearrangement products [45]. This compound could generate 3-deoxyhexosulose compound or intermediate compound via 1,2 enolisation and led to generate several reactive compounds such as formic acid, methylglyoxal and dicarbonyl compound [46, 47]. Strecker aldehydes compound are generated from the Schiff base from

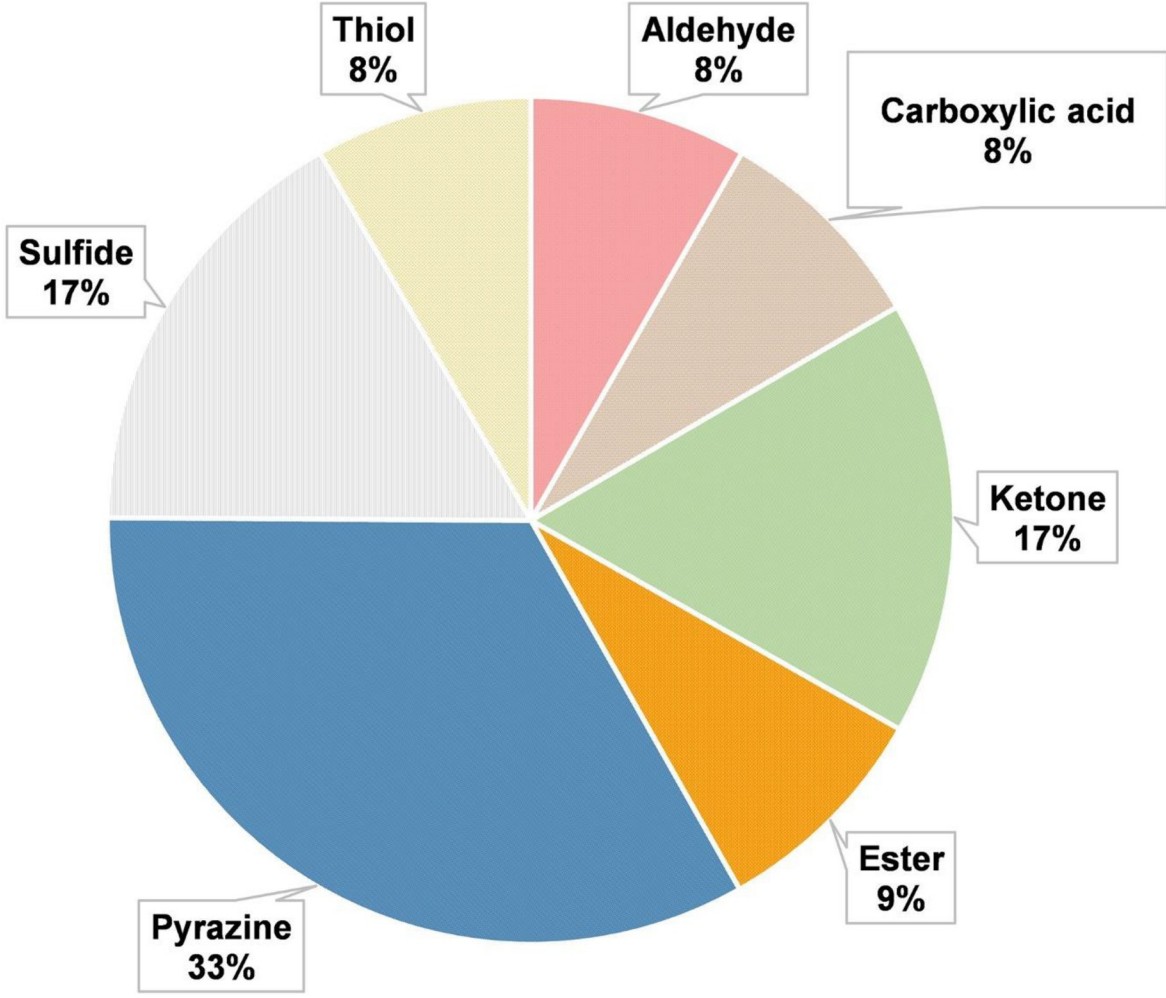

**Fig 2. Distribution of the main classes of volatile compounds found in process flavorings using methionine: dextrose: modified starch: water at the ratio of 56:14:25:5.**

carbonyl -amine condensation of free amino acid with dicarbonyl compounds or reactive compound and followed by decarboxylation in the Maillard reaction [46, 47]. Sasanam et al. [26] identified two Strecker aldehydes, namely 2-methylpropanal and 3-methylbutanal, in process flavorings produced from valine with D-xylose by extrusion process. Granvogl et al. [9] reported the generation of Strecker aldehydes, mainly 2-methylpropanal, when using reducing sugar and valine. According to the authors this compound was produced from oxazolines semi-stable intermediates in a low-moisture system.

Finally, using dextrose as a reducing sugar gave sweet, buttery, creamy, dairy, milky, fatty, cheese, oily and fresh yogurt flavor from ketone compounds, namely, 3-hydroxybutan-2-one and heptane-2,3-dione. Figs 2 and 3 are presented the distribution of the main volatile compounds and flavor wheel obtained for process flavorings from MS25.

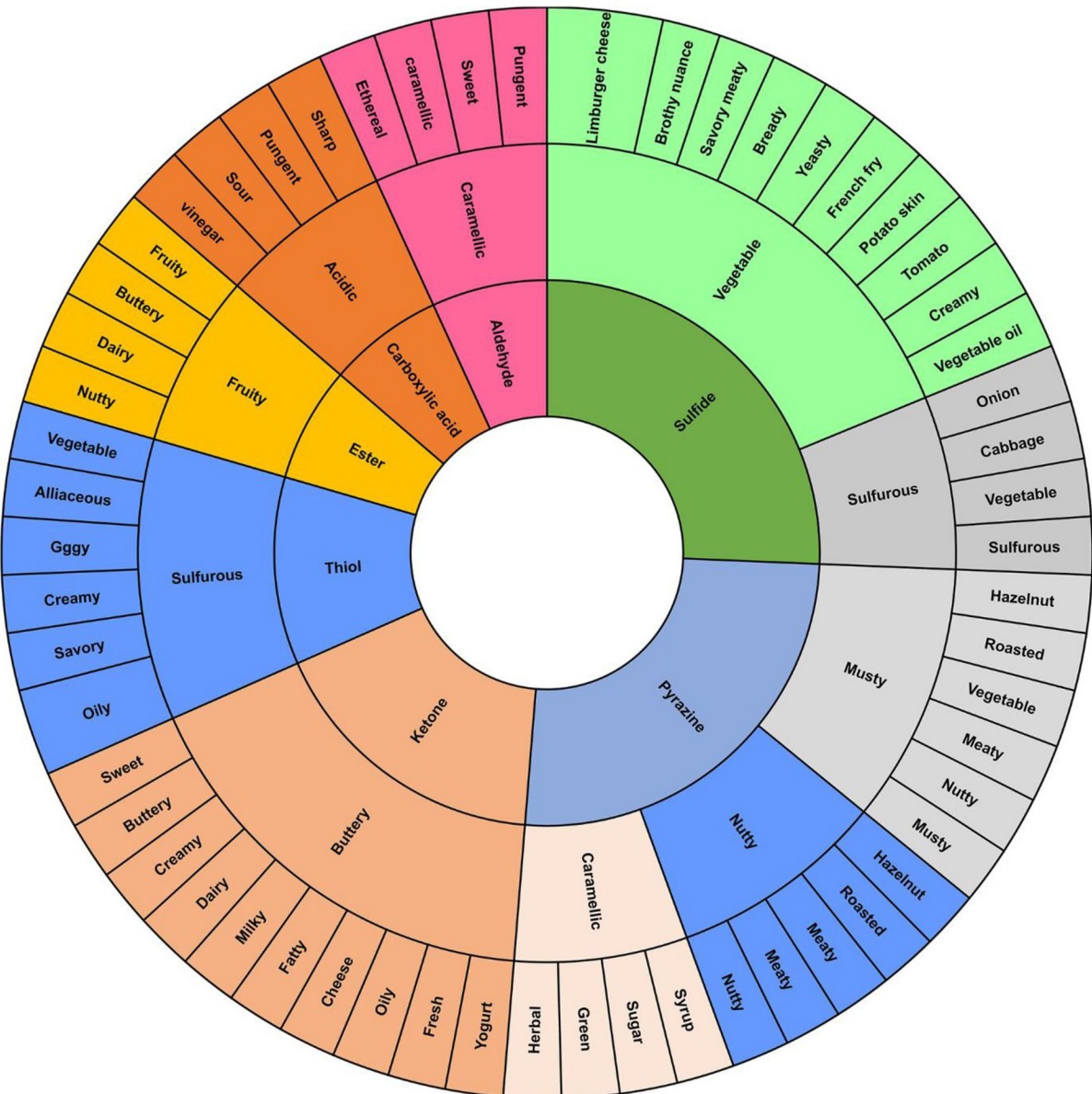

**Fig 3. Flavor wheel of process flavorings extruded from methionine: dextrose: modified starch: water at the ratio of 56:14:25:5.**

## Conclusions

The production of process flavorings using modified starch as a carrier and methionine with dextrose was successfully carried out by direct extrusion. Grinding the extruded process flavorings into powders led to an increase of their $L^*$ value (lightness). The increase of modified starch concentration to 25% allowed reducing the torque during the extrusion from over 85 Nm/cm$^3$ to 10 Nm/cm$^3$, which helped feeding, mixing and cooking along the extruder barrel to occur smoothly with higher output. Some properties of the final product including pH and color indicated the generation of Maillard reaction in the system. Analysis of volatile compounds of the products revealed that sulfurous and thiol compounds, dimethyl disulfide, methional, and methanethiol, were the key meaty volatile compounds present in the produced process flavorings, while pyrazine compounds were detected as the main volatile compounds. These results are new and allow advancing the development of a new technology based on extrusion process for the production of process flavorings. Further studies on the upscale of the extrusion process would be beneficial for the commercial application.

## Acknowledgments

We would like to thank the Research and Researchers for Industries (RRI) scholarship (PHD60I0062) and Mighty International Co., Ltd. for their financial support. Prof. Solange I. Mussatto acknowledges the support from the Novo Nordisk Foundation, Denmark (grant number NNF20SA0066233) and the Thailand Science Research and Innovation Fund (Contract no. KMUTNB-BasicR-65-46).

## Author Contributions

**Conceptualization:** Vilai Rungsardthong.

**Data curation:** Sirinapa Sasanam.

**Funding acquisition:** Sombat Wijuntamook, Vasan Rattananupap.

**Methodology:** Sirinapa Sasanam.

**Project administration:** Benjawan Thumthanaruk, Sombat Wijuntamook, Vasan Rattananupap, Savitri Vatanyoopaisarn.

**Resources:** Chureerat Puttanlek, Dudsadee Uttapap.

**Software:** Sirinapa Sasanam.

**Supervision:** Benjawan Thumthanaruk, Savitri Vatanyoopaisarn, Solange I. Mussatto, Vilai Rungsardthong.

**Validation:** Chureerat Puttanlek, Dudsadee Uttapap.

**Writing – original draft:** Sirinapa Sasanam.

**Writing – review & editing:** Solange I. Mussatto, Vilai Rungsardthong.

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
