## [Decision Letter · Decision Letter 0]

24 Mar 2022

PONE-D-21-38835Extrusion and properties of process flavorings from methionine and dextrose using modified starch as a carrierPLOS ONE

Dear Dr. Sasanam,

Thank you for submitting your manuscript to PLOS ONE. After careful consideration, we feel that it has merit but does not fully meet PLOS ONE’s publication criteria as it currently stands. Therefore, we invite you to submit a revised version of the manuscript that addresses the points raised during the review process.

The reviewers raised several concerns regarding the manuscript quality. English language and the overall quality are below the journal standards. the authors are invited to revise the current version carefully according to reviewers’ comments to meet acceptance criteria.

We look forward to receiving your revised manuscript.

Kind regards,

Walid Elfalleh, Ph.D

Academic Editor

PLOS ONE

Journal Requirements:

We would like to thank the Research and Researchers for Industries (RRI) scholarship (PHD60I0062) and Mighty International Co., Ltd. for their financial support. Prof. Solange I. Mussatto acknowledges the support from the Novo Nordisk Foundation, Denmark (grant number NNF20SA0066233).

The funders play role in the study design,  decision to publish and preparation of the manuscript.

NO authors have competing interests.

6. Please include your tables as part of your main manuscript and remove the individual files. Please note that supplementary tables (should remain/ be uploaded) as separate "supporting information" files.

Reviewers' comments:

Reviewer's Responses to Questions

**Comments to the Author**

1. Is the manuscript technically sound, and do the data support the conclusions?

Reviewer #1: No

Reviewer #2: Yes

Reviewer #3: Yes

2. Has the statistical analysis been performed appropriately and rigorously? 

Reviewer #1: N/A

Reviewer #2: Yes

Reviewer #3: Yes

3. Have the authors made all data underlying the findings in their manuscript fully available?

Reviewer #1: Yes

Reviewer #2: Yes

Reviewer #3: Yes

4. Is the manuscript presented in an intelligible fashion and written in standard English?

Reviewer #1: No

Reviewer #2: Yes

Reviewer #3: Yes

5. Review Comments to the Author

Reviewer #1: The manuscript is highly lacking. English language needs major revision, some sentences make no sense.

There are typos in the entire manuscript. . Write abbreviations in correct manner. Pay attention how the words may be broken in a grammatically correct manner, measuring unity also.

Some sentences in the Introduction are misleading and/or incorrect, while some make no sense. There are no references for some of the statements and the introduction part in general is insubstantial, lacking, superficial, badly written and should be entirely rewritten.

Reviewer #2: The manuscript entitled “Extrusion and properties of process flavourings from methionine and dextrose using modified starch as a carrier” is interesting and has a scope for the journal. However, similar articles are published by the same group of researchers in other journals recently. Need to differentiate this work from the earlier works as this is not a new topic.

Some of the queries or suggestions to improve the manuscript is given below.

Major comments:

1. In color section: Please include the browning index of all the samples and discuss the relationship of Maillard reaction and browning index in detail. Also, discuss how the browning index affects the sensory score of the process flavorings.

2. Sensory evaluation section: What happens to the sensory evaluation of the sample MS45 in comparison with MS25. Is there any effect of torque on the sensory score? Because both samples had the same torque value (10 Nm/cm3).

3. Add the information related to future scope or how your products and findings help the food industry over existing products?

4. All tables, please check the statistical significance representations and analysis, especially the sensory data. Some places (table 3) it’s given as non-significant though there is significant variation in the values given.

5. Line no 134: Is it produced or used?

6. Space is not needed while expressing the ratio. Please do follow the same pattern throughout.

7. Line 252-254: Add reference.

Kindly revise the manuscript incorporating all these suggestions.

Reviewer #3: Overall, this is an interesting paper and can be accepted after revisions are made.

Consider revising title to “Extrusion of process flavorings from methionine and dextrose using

modified starch as a carrier”

Line 29: revise grammar

Line 37: change slightly to slight

Line 102: the explanation of what extrusion is unclear and should be explained further

Line 129-131: The reasoning behind the use of is not clear

Line 207: remove space before 60°C

Line 210: Since the process flavorings exhibited properties suitable for further application as

meaty flavorings, why were the sensory panelists not trained with meaty flavorings. How sure

are the authors that by training the panelist with the five basic tastes they can necessarily identify

meaty flavors?

Line 269: The authors should provide more explanation into what they mean by “resulted in

lower concentration of initial precursor for the Maillard reaction”

Line 284: change nonsignificantly to no significant

Line 301: change volatile to volatiles

6. PLOS authors have the option to publish the peer review history of their article (what does this mean?). If published, this will include your full peer review and any attached files.

Reviewer #1: No

Reviewer #2: No

Reviewer #3: No

---

## [Author Response · Author response to Decision Letter 0]

8 May 2022

Manuscript Number: PONE-D-21-38835

Extrusion and properties of process flavorings from methionine and dextrose using modified starch as a carrier

Academic Editor

Academic Editor Comments to Author: There are no comments.

Dear Academic Editor

Thank you very much for giving us the opportunity to revise the manuscript. I also would like to express my sincere thanks to the reviewers for giving us comments and suggestions. The manuscript has been revised and improved as kindly suggested. Attached please find the revised manuscript and the responses to each reviewer.

Looking forward to the acceptance of the paper.

Best regards,

Vilai Rungsardthong

Referee(s)' Comments to Author:

Reviewer #1: The manuscript is highly lacking. English language needs major revision, some sentences make no sense.

There are typos in the entire manuscript. Write abbreviations in correct manner. Pay attention how the words may be broken in a grammatically correct manner, measuring unity also.

Some sentences in the Introduction are misleading and/or incorrect, while some make no sense. There are no references for some of the statements and the introduction part in general is insubstantial, lacking, superficial, badly written and should be entirely rewritten.

Ans: Thank you very much for your comments. The manuscript was improved and rewritten point by point as detailed in the attached file. 

Reviewer #2: The manuscript entitled “Extrusion and properties of process flavourings from methionine and dextrose using modified starch as a carrier” is interesting and has a scope for the journal. However, similar articles are published by the same group of researchers in other journals recently. Need to differentiate this work from the earlier works as this is not a new topic.

Some of the queries or suggestions to improve the manuscript is given below.

Ans: Thank you very much for spending your time looking through the manuscript and giving us the insightful comments and valuable suggestions. Though similar articles were published by our group recently, but the scope and content of this work is different from our previous works. The scopes and objectives of our previously published works are described as follows. 

1) The first paper, Sasanam, S., Rungsardthong, V., Thumthanaruk, B., Wijuntamook, S., Rattananupap, V., Vatanyoopaisarn, S., Puttanlek, C., Uttapap, D., and Mussatto, S.I. (2021). Properties and volatile profile of process flavorings prepared from D-xylose with glycine, alanine or valine by direct extrusion method. Food Bioscience, 44, 101371. This study investigated the production of process flavorings by direct extrusion method using a model system consisted of 80% (w/w) D-xylose and glycine, alanine, or valine at 20% (w/w). The extruder’s barrel temperatures were controlled at 65, 80, and 50 °C. The aroma compounds in the obtained products were analyzed by gas chromatography-mass-spectrometry (GC-MS). In addition, some properties relevant for food application including solubility, water absorption index, and oil absorption capacity were also determined.

2) Sasanam, S., Rungsardthong, V., Thumthanaruk, B., Wijuntamook, S., Rattananupap, V., Vatanyoopaisarn, S., Puttanlek, C., Uttapap, D., and Mussatto, S.I. (2022). Production of process flavorings from methionine, thiamine with d-xylose or dextrose by direct extrusion: Physical properties and volatile profiles. Journal of Food Science, 87(3), 895-910. This research aimed to evaluate the use of methionine, thiamine and reducing sugars to develop process flavorings by direct extrusion, as a potential alternative to the conventional method. The mixed substrates consisted of methionine: D-xylose (MX), methionine: dextrose (MD), thiamine: D-xylose (TX), and thiamine: dextrose (TD) at 80:20 by weight.

We had found some difficulties (high toque during the extrusion) in extruding the process flavorings from methionine and dextrose or glucose from the work described in the second paper. Modified starch was not used in our previous works. Consequently, in this work we had tried to solve the high torque problem occurred previously by using a modified starch as a carrier to facilitate the production process. In conclusion, this paper describes further study from the problem occurred in our previous works. The scopes and objectives were described in the “Introduction”.

In addition, we have checked, revised, and improved the manuscript point by point as kindly suggested as following details.

Major comments:

1. In color section: Please include the browning index of all the samples and discuss the relationship of Maillard reaction and browning index in detail. Also, discuss how the browning index affects the sensory score of the process flavorings.

We have added the calculation of the browning index in the method as below.

The browning index (BI) of each process flavorings was calculated based on the CIE values of L*, a*, and b* [28] as presented in the following equation.

BI = 100 x (X – 0.31)/ 0.17

 Where, X = (a* + 1.75 L*) / (5.645L* + a* - 3.012b*) 

Please see: Line 180-184, Page 8 and Table 2 

We have added more discussions in the manuscript. “The browning index (BI) is an important parameter to measure the brown color of process flavorings produced from the Maillard reaction [38]. The BI of each sample are presented in Table 2. After grinding, BI of all process flavorings tended to significantly decrease (p< 0.05). This could be due to the higher light scattering occurred when the extrudates were size reduced into finer particles. The BI of powdered extrudates of all formulations ranged 11.92- 38.12. However, the BI of each sample did not correlate well with the sensory acceptability in term of color attribute as presented in Table 3. The sensory panelists preferred the product color from the samples extruded with the modified starch 15-45% than the use of 5% modified starch, but the difference was very slight. However, all samples from the modified starch 5-45% obtained non-significant difference for overall acceptability (5.53 – 6.00).” 

Please see: Line 290-300, Page 12-13

We also have added these two references in the reference lists 

28. Mohapatra D, Mishra S, Sutar N. 2010. Banana and its by-product utilization: an overview. J. Sci. Ind. Res. 2010; 69: 323–329. 

38. Zambrano-Zaragoza ML, Mercado-Silva E, Gutiérrez-Cortez E, Cornejo-Villegas M A, Quintanar-Guerrero D. The effect of nano-coatings with α-tocopherol and xanthan gum on shelf-life and browning index of fresh-cut “Red Delicious” apples. Innov Food Sci Emerg Technol. 2014; 22: 188-196. doi.10.1016/j.ifset.2013.09.008

Please see: Line 526-527 and 555-558, page 22-23 

Table 2. Color indicators (L*, a*, b*) of the extruded process flavorings produced from various ratios of substrates, before and after size reduction into powder. 

Sample Methionine: dextrose:

modified starch:

water Extrudates Powdered extrudates

 L* a* b* BI1 L* a* b* BI2

MS5 72:18:5:5 79.33 ± 2.51a 6.00 ± 0.00c 33.33 ± 2.51bc 60.09 ± 5.11d 89.00 ± 2.64ab 3.00 ± 0.00a 24.66 ± 0.57b 34.26 ± 0.89b

MS15 64:16:15:5 71.66 ± 4.50b 6.00 ± 2.00a 36.00 ± 2.64ab 80.19 ± 14.22b 90.33 ± 3.05ab 1.66 ± 0.57bc 28.33 ± 1.52a 38.12 ± 1.88a

MS25 56:14:25:5 68.00 ± 5.00b 9.33 ± 2.08ab 34.00 ± 1.00b 99.12 ± 42.76a 87.00 ± 2.00b 1.00 ± 0.00d 22.00 ± 1.00c 29.36 ± 2.03c

MS35 42:12:35:5 83.00 ± 6.92a 6.66 ± 0.57bc 28.00 ± 2.00d 55.96 ± 8.43e 87.66 ± 3.05b 2.33 ± 0.57ab 9.66 ± 0.57d 13.38 ± 1.28d

MS45 40:10:45:5 88.00 ± 2.14a 7.33 ± 1.52abc 37.00 ± 4.00a 73.64 ± 22.55c 93.00 ± 2.57a 1.00 ± 1.00cd 9.33 ± 0.57d 11.92 ± 1.68d

Means ± SD in the same column, followed by the same letter, are not significantly different at p<0.05. 

BI1 = Browning index of extrudates 

BI2 = Browning index of powdered extrudates 

2. Sensory evaluation section: What happens to the sensory evaluation of the sample MS45 in comparison with MS25. Is there any effect of torque on the sensory score? Because both samples had the same torque value (10 Nm/cm3).

Ans: Thank you very much for your insight question. The results showed that the addition of higher modified starch could facilitate the extrusion with lower torque, for example for the MS25 and MS45. Running the samples with higher ratio of the modified starch tended to exhibit lower torque during the extrusion and yielded samples with lower BI. However, both samples still obtained non-significant difference for color preference. Increasing the ratio of the modified starch resulted to a lower concentration of methionine and dextrose in the sample mix thus a lower concentration of the brown products from the Maillard reaction was generated. 

We have added “The results showed that the addition of higher modified starch could facilitate the extrusion with lower torque, for example for the MS25 and MS45. Running the samples with higher ratio of the modified starch tended to exhibit lower torque during the extrusion and yielded samples with lower BI. However, both samples still obtained non-significant difference for color preference. Increasing the ratio of the modified starch resulted to a lower concentration of methionine and dextrose in the sample mix thus a lower concentration of the brown products from the Maillard reaction was generated.” in the manuscript. 

Please see: Line 310-316, Page 13

3. Add the information related to future scope or how your products and findings help the food industry over existing products?

Ans: We have added more information related to future investigation from our findings. Thank you very much

“The findings from our work would be potentially help the food industry to use the extrusion to produce process flavorings. The use of carriers including modified starches can facilitate the extrusion to be controlled more efficiently. In addition, modification of the type and concentration of sugars, amino acids, and carriers would be beneficial for further applications.” 

Please see: Line 322-326, Page 14

4. All tables, please check the statistical significance representations and analysis, especially the sensory data. Some places (table 3) it’s given as non-significant though there is significant variation in the values given.

Ans: Thank you very much for your comments. We have carefully checked the statistical analysis and found that they are correctly presented. Though the data in Table 3 seemed to be significantly different, they are mostly non-significantly different based on statistical analysis. This could be because of the high SD of each result. We have attached the analysis data by SPSS as follows: 

Results in Table 3

Table 3. Sensory evaluation of process flavorings produced from various ratios of mixed substrates using a 9-point scoring scale.

Sample 

Methionine: dextrose:

modified starch: water Aromans Taste ns Color Water solubilityns Overall acceptabilityns

MS5 72:18:5:5 5.40 ± 2.35 5.13 ± 1.99 5.20 ± 1.74b 6.60 ± 1.12 5.53 ± 1.76

MS15 64:16:15:5 5.66 ± 1.67 5.06 ± 1.94 6.80 ± 1.37a 6.80 ± 1.01 5.86 ± 1.64

MS25 56:14:25:5 5.46 ± 1.68 6.20 ± 0.07 5.60 ± 1.68ab 7.00 ± 0.84 6.00 ± 1.25

MS35 42:12:35:5

 5.26 ± 1.94 5.40 ± 1.95 6.00 ± 1.81a 6.20 ± 1.26 5.66 ± 1.67

MS45 40:10:45:5 5.00 ± 1.64 5.80 ± 1.20 5.60 ± 1.80ab 7.00 ± 0.65 5.66 ± 1.44

The temperature in zones 1, 2 was controlled at 100, 120 °C with a screw speed of 30 rpm.

Means ± SD in the same column, followed by the same letter, are not significantly different at p<0.05. 

ns: Not significantly different at p<0.05 

Output from the statistical program (SPSS)

Descriptives

Aroma

 N Mean Std. Deviation Std. Error 95% Confidence Interval for Mean Minimum Maximum

 Lower Bound Upper Bound 

1.00 (MS5) 15 5.4000 2.35433 .60788 4.0962 6.7038 1.00 8.00

2.00 (MS15) 15 5.6667 1.67616 .43278 4.7384 6.5949 3.00 8.00

3.00 (MS25) 15 5.4667 1.68466 .43498 4.5337 6.3996 2.00 9.00

4.00 (MS35) 15 5.2667 1.94447 .50206 4.1899 6.3435 1.00 8.00

5.00 (MS45) 15 5.0000 1.64751 .42538 4.0876 5.9124 2.00 7.00

Total 75 5.3600 1.84274 .21278 4.9360 5.7840 1.00 9.00

Homogeneous Subsets

Aroma

 Aroma N Subset for alpha = 0.05

 1

Duncana dimension1 5.00 15 5.0000

 4.00 15 5.2667

 1.00 15 5.4000

 3.00 15 5.4667

 2.00 15 5.6667

 Sig. .397

Means for groups in homogeneous subsets are displayed.

a. Uses Harmonic Mean Sample Size = 15.000.

Descriptives

Taste

 N Mean Std. Deviation Std. Error 95% Confidence Interval for Mean Minimum Maximum

 Lower Bound Upper Bound 

1.00 (MS5) 15 5.1333 1.99523 .51517 4.0284 6.2383 2.00 8.00

2.00 (MS15) 15 5.0667 1.94447 .50206 3.9899 6.1435 2.00 8.00

3.00 (MS25) 15 6.2000 .77460 .20000 5.7710 6.6290 5.00 8.00

4.00 (MS35) 15 5.4000 1.95667 .50521 4.3164 6.4836 1.00 8.00

5.00 (MS45) 15 5.8000 1.20712 .31168 5.1315 6.4685 4.00 8.00

Total 75 5.5200 1.66328 .19206 5.1373 5.9027 1.00 8.00

Homogeneous Subsets

 Taste

 Taste N Subset for alpha = 0.05

 1

Duncana dimension1 2.00 15 5.0667

 1.00 15 5.1333

 4.00 15 5.4000

 5.00 15 5.8000

 3.00 15 6.2000

 Sig. .098

 Means for groups in homogeneous subsets are displayed.

 a. Uses Harmonic Mean Sample Size = 15.000.

Descriptives

Color

 N Mean Std. Deviation Std. Error 95% Confidence Interval for Mean Minimum Maximum

 Lower Bound Upper Bound 

1.00 (MS5) 15 5.2000 1.74028 .44934 4.2363 6.1637 2.00 7.00

2.00 (MS15) 15 6.8000 1.37321 .35456 6.0395 7.5605 5.00 9.00

3.00 (MS25) 15 5.6000 1.68184 .43425 4.6686 6.5314 1.00 8.00

4.00 (MS35) 15 6.0000 1.81265 .46803 4.9962 7.0038 2.00 8.00

5.00 (MS45) 15 5.6000 1.80476 .46599 4.6006 6.5994 2.00 8.00

Total 75 5.8400 1.73236 .20004 5.4414 6.2386 1.00 9.00

Homogeneous Subsets

 Color

 Color N Subset for alpha = 0.05

 1 2

Duncana dimension1 1.00 15 5.2000 

 3.00 15 5.6000 5.6000

 5.00 15 5.6000 5.6000

 4.00 15 6.0000 6.0000

 2.00 15 6.8000

 Sig. .244 .079

 Means for groups in homogeneous subsets are displayed.

 a. Uses Harmonic Mean Sample Size = 15.000.

Descriptives

Water solubility

 N Mean Std. Deviation Std. Error 95% Confidence Interval for Mean Minimum Maximum

 Lower Bound Upper Bound 

1.00 (MS5) 15 6.6000 1.12122 .28950 5.9791 7.2209 4.00 8.00

2.00 (MS15) 15 6.8000 1.01419 .26186 6.2384 7.3616 4.00 8.00

3.00 (MS25) 15 7.0000 .84515 .21822 6.5320 7.4680 5.00 8.00

4.00 (MS35) 15 6.2000 1.26491 .32660 5.4995 6.9005 4.00 8.00

5.00 (MS45) 15 7.0000 .65465 .16903 6.6375 7.3625 6.00 8.00

Total 75 6.7200 1.02086 .11788 6.4851 6.9549 4.00 8.00

Homogeneous Subsets

Water solubility

 Water solubility N Subset for alpha = 0.05

 1

Duncana dimension1 4.00 15 6.2000

 1.00 15 6.6000

 2.00 15 6.8000

 3.00 15 7.0000

 5.00 15 7.0000

 Sig. .054

Means for groups in homogeneous subsets are displayed.

a. Uses Harmonic Mean Sample Size = 15.000.

Descriptives

Overall acceptability

 N Mean Std. Deviation Std. Error 95% Confidence Interval for Mean Minimum Maximum

 Lower Bound Upper Bound 

1.00 (MS5) 15 5.5333 1.76743 .45635 4.5546 6.5121 2.00 8.00

2.00 (MS15) 15 5.8667 1.64172 .42389 4.9575 6.7758 3.00 9.00

3.00 (MS25) 15 6.0000 1.25357 .32367 5.3058 6.6942 4.00 8.00

4.00 (MS35) 15 5.6667 1.67616 .43278 4.7384 6.5949 2.00 8.00

5.00 (MS45) 15 5.6667 1.44749 .37374 4.8651 6.4683 4.00 8.00

Total 75 5.7467 1.53423 .17716 5.3937 6.0997 2.00 9.00

Homogeneous Subsets

Overall acceptability

 Overall acceptability N Subset for alpha = 0.05

 1

Duncana dimension1 1.00 15 5.5333

 4.00 15 5.6667

 5.00 15 5.6667

 2.00 15 5.8667

 3.00 15 6.0000

 Sig. .477

Means for groups in homogeneous subsets are displayed.

a. Uses Harmonic Mean Sample Size = 15.000.

5. Line no 134: Is it produced or used?

Ans: Edited as suggested. Thank you very much.

Please see: Line 136, Page 6 

6. Space is not needed while expressing the ratio. Please do follow the same pattern throughout.

Ans: Edited as kindly suggested. Thank you very much.

7. Line 252-254: Add reference.

Kindly revise the manuscript incorporating all these suggestions.

Ans: We have added the reference in the text and reference lists as kindly suggested. Thank you very much.

Reference in text

However, the role of the amino acid in the Maillard reaction could be separated into two parts: the first part is to promote a sugar-amino condensation, while the second part is to generate specific aromas via the Strecker degradation [26].

Please see: Line 259-262, Page 11

Reference in reference lists

26. Sasanam S, Rungsardthong V, Thumthanaruk B, Wijuntamook S, Rattananupap V, Wijuntamook S, et al. Properties and volatile profile of process flavorings prepared from D-xylose with glycine, alanine or valine by direct extrusion method. Food Biosci. 2021a: 101371. doi.10.1016/j.fbio.2021.101371.

Please see: Line 521-524, Page 22

Reviewer #3: Overall, this is an interesting paper and can be accepted after revisions are made.

Ans: Thank you very much for your kind comments. We have checked, revised, and improved the manuscript point by point as kindly suggested as detailed following.

1.Consider revising title to “Extrusion of process flavorings from methionine and dextrose using modified starch as a carrier”

Ans: Thank you very much. We have revised the title from “Extrusion and properties of process flavorings from methionine and dextrose using modified starch as a carrier” to “Extrusion of process flavorings from methionine and dextrose using modified starch as a carrier” as kindly suggested.

Please see: Title (line 1-2), Page 1

2. Line 29: revise grammar

Ans: We have edited the sentence and added “Five formulations of process flavorings with different ratios of methionine: dextrose: modified starch: water as MS5 (72:18:5:5), MS15 (64:16:15:5), MS25 (56:14:25:5), MS35 (42:12:35:5), and MS45 (40:10:45:5) were prepared and feded into the extruder.” in the manuscript. Thank you very much.

Please see: Abstract, Line 28-31, Page 2

3. Line 37: change slightly to slight

Ans: Edited Thank you very much.

Please see: Line 37, page 2

4. Line 102: the explanation of what extrusion is unclear and should be explained further

Ans: we have revised and explained about what extrusion as following:

 “Extrusion process is a potential modern, green and environment-friendly technology that combines several processing steps such as mixing, cooking, sterilization, heating, and forming.” Thank you very much.

Please see: Line 102-104, Page 5

5. Line 129-131: The reasoning behind the use of is not clear

Ans: We have revised and added “However, to the best of our knowledge, there are no reports on the use of modified cassava starch as a carrier to solve the problems related to the high torque, too high viscosity, and stickiness of the raw materials to produce process flavorings by extrusion.” Thank you very much.

Please see: Line 128-131, Page 6

6. Line 207: remove space before 60°C

Ans: Edited. Thank you very much.

Please see: Line 216, Page 9

7. Line 210: Since the process flavorings exhibited properties suitable for further application as meaty flavorings, why were the sensory panelists not trained with meaty flavorings. How sure are the authors that by training the panelist with the five basic tastes they can necessarily identify meaty flavors?

Ans: We appreciate the reviewer’s insightful comments. All trained panelists are specialists in our collaborated flavor company who has long experiences in the production and quality control for the process flavorings (both for local consumption and for the export to many countries). They were trained for five basic tastes as well as for the meaty flavorings since meaty flavorings are one of the main products from this company. The company have a standard program training for the panelists in their workplace (confidential data). Therefore, we have confidence that the trained panelists are suitable for further application in meaty flavorings in the future.

8. Line 269: The authors should provide more explanation into what they mean by “resulted in lower concentration of initial precursor for the Maillard reaction”

Ans: We have revised the sentence to “Using high ratio of the modified starch from 5% to 45% led to a decrease in the concentration of methionine and dextrose which were the initial precursors for the Maillard reaction in the system.” in the manuscript.

Please see: Line 276-278, page 12

9.Line 284: change nonsignificantly to no significant

Ans: Revised. Thank you very much. 

Please see: Line 304, page 13

10. Line 301: change volatile to volatiles

Ans: Edited. Thank you very much.

Please see: Line 332, page 14

---

## [Decision Letter · Decision Letter 1]

30 May 2022

Extrusion of process flavorings from methionine and dextrose using modified starch as a carrier

PONE-D-21-38835R1

Dear Sirinapa Sasanam,

We’re pleased to inform you that your manuscript has been judged scientifically suitable for publication and will be formally accepted for publication once it meets all outstanding technical requirements.

Kind regards,

Muhammad Sajid Arshad

Academic Editor

PLOS ONE

Additional Editor Comments (optional):

Reviewers' comments:

Reviewer's Responses to Questions

**Comments to the Author**

1. If the authors have adequately addressed your comments raised in a previous round of review and you feel that this manuscript is now acceptable for publication, you may indicate that here to bypass the “Comments to the Author” section, enter your conflict of interest statement in the “Confidential to Editor” section, and submit your "Accept" recommendation.

Reviewer #1: All comments have been addressed

Reviewer #3: All comments have been addressed

2. Is the manuscript technically sound, and do the data support the conclusions?

Reviewer #1: Yes

Reviewer #3: Yes

3. Has the statistical analysis been performed appropriately and rigorously? 

Reviewer #1: Yes

Reviewer #3: Yes

4. Have the authors made all data underlying the findings in their manuscript fully available?

Reviewer #1: Yes

Reviewer #3: Yes

5. Is the manuscript presented in an intelligible fashion and written in standard English?

Reviewer #1: Yes

Reviewer #3: Yes

6. Review Comments to the Author

Reviewer #1: The manuscript was revised and re-written taking into consideration the reviewer’s suggestions and comments. The current version has been improved. Therefore, It is suitable for publication.

Reviewer #3: All revisions appropriately done. Paper is now very interesting to read and should be accepted for publication

7. PLOS authors have the option to publish the peer review history of their article (what does this mean?). If published, this will include your full peer review and any attached files.

Reviewer #1: No

Reviewer #3: No

---

## [Editor Report · Acceptance letter]

17 Jun 2022

PONE-D-21-38835R1 

Extrusion of process flavorings from methionine and dextrose using modified starch as a carrier 

Dear Dr. Sasanam:

I'm pleased to inform you that your manuscript has been deemed suitable for publication in PLOS ONE. Congratulations! Your manuscript is now with our production department. 

Kind regards, 

on behalf of

Dr. Muhammad Sajid Arshad 

Academic Editor

PLOS ONE